# Linkage Analysis between Finance and Environmental Protection Sectors in China: An Approach to Evaluating Green Finance

**DOI:** 10.3390/ijerph18052634

**Published:** 2021-03-05

**Authors:** Libo Li, Wenbing Wu, Mingyu Zhang, Lu Lin

**Affiliations:** 1School of Economics and Management, Beijing Jiaotong University, Beijing 100044, China; lilibo@abchina.com (L.L.); myzhang@bjtu.edu.cn (M.Z.); 2School of Economics and Management, China University of Petroleum Beijing, Beijing 102249, China; linlu26@hotmail.com

**Keywords:** green finance, environmental protection sector, industrial linkage analysis, input–output analysis

## Abstract

Given the growing awareness of sustainable development, the environmental protection industry has attracted much attention. Green finance has developed rapidly in policymaking and practices. This study provides a framework for evaluating green finance via linkage analysis based on input–output theory. Measurements on industrial linkages are calculated in China in two provinces from 2002 to 2018, which study the relationship between finance and environmental protection sectors. The results show that the environmental protection sector (EPS) in China has gradually developed from a sector with weak backward and strong forward linkages to a sector with strong backward and weak forward linkages from 2002 to 2015; however, in 2017 and 2018, the EPS returned to a sector with weak backward and strong forward linkages. At the provincial level, the EPS used to be a key sector with strong backward and forward linkages. The connection between the finance sector and the EPS rose first, then declined in the country and the Zhejiang province; Guangdong had a similar evolution in the former period, but it had a rising trend in the latest year. The findings provide insights for further promoting the support from the finance sector to the environmental protection activities.

## 1. Introduction

With growing concern for global climate change, environmental pollution issues, and natural resource depletion, more governments, organizations, and the public realize the importance of climate change mitigation, pollution prevention and control, and sustainable resource utilization. The demands for the environmental protection industry, i.e., products to prevent, control, measure, or minimize environmental pollution and resource depletion, have increased [1]. The environmental protection industry is related to other industries in the economy from two aspects: providing environmental goods and services necessary for most economic activities and using various products and services from other industries. For example, sewage treatment plants collect effluent from industrial and municipal sources and treat it to a purification level that enables its reuse in agricultural and industrial industries. Meanwhile, sewage treatment plants should use treatment chemicals produced by the manufacturing of chemicals. In other words, the environmental protection industry has essential effects on the economy because of its direct or indirect relationships with other industries. In the context of globalization and technology innovation, governments are aware of the potential of the environmental protection industry in stimulating economic growth, creating jobs, and playing a significant role in the economic transition towards sustainable development. The global market of the environmental industry is expected to rise to US$1.9 trillion by 2020 [2]. In developing countries, given the growing awareness of sustainable development, the demand for environmental goods and services has increased rapidly. In China, the environmental protection industry has been increasing since 2016 by an annual rate of more than 10% in operating revenue and 3.9% in profit. The environmental protection industry’s direct contribution rate to the national economy increased from 0.3% in 2004 to 3.1% in 2019 [3].

There should be a massive demand for investment to develop environmental protection projects. The investment and financing of environmental protection projects are generally characterized by a long investment cycle and low return, making it difficult to attract enough capital from financial markets. In the past, governments’ investment accounted for the major capital sources of the environmental protection industry. In recent years, governments have attempted to release policies and plans to promote financial resources going into this industry, which is a significant part of green finance. Consequently, the approaches of investment in the environmental protection industry have increased. The environmental projects heavily rely on government financial transfer payments and bank loans. However, various market-oriented financing approaches have emerged, such as public–private partnership (PPP) mode, funds for green industries, and environmental rights trading. Green finance systems that combine governments, financial institutes, other investors, and suppliers and purchasers of environmental goods and services have been formed [4].

China is the second largest economy in the world. Its highly rapid economic development has lasted for more than 30 years. In the early stage of rapid economic development, environmental pollution and resource depletion issues were ignored. Fortunately, environmental protection and resource recovery have attracted more attention. China’s government proposes transforming economic development to a more environmentally- friendly, resource-saving, and low-carbon target. In this transformation process, to build a green financial system would be an important approach. The development of green finance in China has developed sharply. Some statistical data proves this rapid development of green finance in China. The issuance scale of green bonds reached US$257.7 billion in 2019, an increase of 51.06% over that in 2018, ranking second on the global green bond issuance scale [5]. The amount of green loan balance was 10.22 trillion Yuan, accounting for more than 10% of the total loan balance [5]. In addition to a large number of green financial products, China provides many important lessons and experiences in developing a green financial system. Therefore, the evaluation of its current trends is needed, which will be helpful in scientific policymaking and future development

This study aims to investigate the development situation of the environmental protection industry and its support from the finance sector in China. Based on the input–output theory about industrial relationships, this study uses linkage analysis to quantify the role of the environmental protection industry in the economic system and the relationship between the environmental protection industry and finance industry. By reviewing the existing literature, although the previous studies concerning green finance in China have developed during the past several years, this study presents its contributions in two aspects. Firstly, from the theoretical level, few studies have focused on a comprehensive view of financial supports to environmental protection sectors. Previous studies mainly concerned different green finance products separately, such as green credit, green bond, green funding, or carbon trading. This study provides a macroeconomic approach in quantifying the development of green finance. This approach is based on the input–output (I–O) theory, which can evaluate economic activities comprehensively and systematically. Secondly, regarding an empirical perspective, this approach is easy to understand and handle for policymakers and researchers. Meanwhile, the data basis of this approach is the public I–O tables, which are available and accessible. Therefore, this approach can be extended to different spatial units that have local I–O tables.

The rest of this study is organized as follows. The following section summarizes the literature on green finance and industrial linkage. Section 3 introduces the I–O method, linkage measurements and the data used. Section 4 presents the major results in linkage effects and annual variations of the EPS, as well as the inter-sectoral linkages between the EPS and finance sectors. Section 5 proposes some policy implications based on result discussion. Section 6 concludes some remarkable findings.

## 2. Literature Review

### 2.1. Green Finance

Along with extensive research on climate change, green finance has attracted more attention from academic scholars and policymakers. The definitions of green finance are various. The definition of green finance occurred in 1990s, when Devas (1994) proposed this definition to describe the impact of environmental factors on the financial sector [6]. Salazar (1998) asserted that green finance is the channel connecting the finance industry to the environmental protection industry [7]. The G20 defines green finance as the funding of investments that provide environmental benefits, including environmental pollution reduction, natural resource depletion, and climate change mitigation [8,9]. In China, the national policy’s authoritative definition has a similar meaning as the G20′s. In China, green finance promotion can be traced back to the 1990s. However, its booming development occurred in the 2010s, especially when an official policy of creating the Chinese national green finance system was released in 2016 [10,11].

In the context of the rapid development of green finance, most green finance research has focused on the concept of green finance, its theoretical framework, and its development path and mechanism. Scholtens discussed the relationship between finance and enterprise social responsibility [12]. Taghizadeh-Hesary and Yoshino discussed the theoretical model for inducing private participation in green finance and investment. The authors proposed that green credit guarantee schemes and distributed ledger technologies could encourage the private sector to invest in environmental projects [13]. D’Orazio and Popoyan discussed the function of promoting green finance of macroprudential tools in different countries [14]. Secinaro et al. investigated the impact of climate change mitigation policies on corporate financial performance. The authors found that the firms employed environmental practices to reduce environmental risks, and then lowered production costs and increased profits [15]. Other scholars focused on some green finance products, like green credit, green bond, and green funding, and investigate their release status, innovations, and effectiveness in promoting green economic development.

To understand its development situations, scholars attempt to assess and evaluate green finance quantitatively and qualitatively. Zhang et al. analyzed the present situation of green finance and proposed its development trend using the bibliometric analysis method [6]. Wang et al. evaluated the development levels of provincial green finance in three northern provinces in China, i.e., Beijing, Tianjin, and Hebei, from 2007 to 2018. The entropy method and DEA-Malmquist were used to assess the relative development levels and input–output efficiencies of green finance in the three provinces, respectively [16]. Using text mining technology, Ren analyzed green finance policies and news, and proposed an index to assess green finance development in China [17]. Based on the data of 1040 Chinese listing companies, Zhang et al. evaluated green finance’s development situations at the provincial level [18].

As mentioned above, scholars mainly used the entropy method, analytic hierarchy process, factor analysis, and principal component analysis to establish index systems to assess the development of green finance. Some scholars used the data from questionnaire surveys and interviews to summarize the situations and attitudes of financial institutions in the development of green finance [18,19,20]. These studies do not directly quantify the support for green economic activities from the financial industry, but measure the development of green finance through some alternative indicators, such as the proportion of green credit, the scale of green bond issuance, and the number of clean development mechanism (CDM) projects. The selection and alternative indicators and the mathematic methods dealing with them would greatly impact the assessment results. On the other hand, these methods are difficult to extend to other regions or other financial institutions due to the limitation of data availability.

To summarize, the previous studies have reported a widely accepted definition of green finance and highlighted the need for evaluating its development situations. However, the studies reviewed here assess the status of green finance using indirect indices because of the lack of systematic data on green investment. For example, in the Chinese statistical system of green credit, only the green credits in 21 commercial banks are collected. Another potential problem is that the development situation results mainly rely on the indices selected, which might change along with case locations. Therefore, it is necessary to establish an approach to evaluate green finance development more broadly and in a way that is less data-consuming. The study by Ling et al. provided this kind of approach. In the input–output analysis framework, the authors disaggregated environmental protection sectors from the classical sectors and compiled new input–output tables. The authors then measured the relationship between the environmental protection and finance sectors using the linkage analysis method based on new tables at the national level [21].

### 2.2. Industrial Linkage

From the perspective of industrial relationships, evaluating the development level of green finance is essential to explore the strength of the inter-connections between the financial industry and environmental protection industry. Inter-sectoral linkages reflect the inter-connections between the sectors in an economy. The inter-sectoral linkages involve backward and forward linkages. In the I–O framework, production by a particular sector has two effects on other sectors in the economy [22]. One is to sell its products to other sectors as a supplier, and another one is to purchase products by other sectors as a purchaser. The former means that if the given sector increases its outputs, there will be increased supplies for other sectors that use products by the given sector. This inter-connection of the given sector with the downstream sectors is called forward linkage. On the other hand, the latter effect indicates that if the given sector increases its outputs, there will be increased demands on the other sectors whose products are used as inputs in production by the given sector. This inter-connection of the given sector with the upstream sectors is defined as backward linkage.

The concept of linkage was proposed by Hirschman [23] and Rasmussen [24] in the 1950s. There are several measurements proposed to assess inter-sectoral linkages. Wen and Wang [25] divided the measurement methods into four types: the multiplier method, sensitivity analyses, the hypothetical extraction method, and the modified hypothetical extraction method. Cai and Leung [26] mentioned two methods of the linkages, the traditional method using direct and total linkage measurement indices and the hypothetical extraction method. The linkage analysis plays a vital role in many aspects of economic development. The information by linkage analysis is valuable in identifying key sectors and forecasting the economic activities [27], because the relative size of linkage metrics can reflect strengths, weaknesses, and vulnerabilities of the particular sector within the economy. The industrial linkages provide an approach to diffusing knowledge and new technologies; therefore, the linkage analysis is an important technological innovation analysis [28]. The environmental and carbon emission extension I–O tables and their linkage analysis were implemented to evaluate environmental and carbon impacts by key sectors and help policymaking in pollution and carbon reduction [25,29,30].

The measurement indices of linkages can be classified into two categories. One refers to assessing the inter-connection in the economy of a particular sector [31,32,33,34]. For example, Fan et al. [35] used backward and forward correlation coefficients to calculate the effects of the environmental protection industry on other industries and the contribution to the economy. The other is to measure the strength of inter-dependency between two sectors [27]. Song et al. [27] used the hypothetical extraction method to assess the relationship between real estate and construction. In this study, we use the former indices to assess the role of the EPS in China’s economy, then use the latter indices to analyze the relationship between the EPS and finance sectors.

## 3. Methods and Data

### 3.1. Methods: Linkage Analysis Based on the Input–Output Model

#### 3.1.1. General Framework of the Input–Output Model

The linkages between the finance sector and environmental protection sector (EPS) can be used to evaluate the development level of green finance. The strength and changing trends of the linkages between both sectors are suitable for assessing what finance sector supports EPS. In this study, the linkage analysis is based on an input–output model, a top-down macroeconomic technique using sectoral interdependency [30,36]. The input–output (I–O) model is widely used for quantifying sectoral linkages mainly from three aspects. First, scholars have used the I–O model to describe the interdependency among the economic sectors, including direct, indirect, and total linkages [30,36]. Second, the I–O model is used to examine the effects on an entire economy of a given sector. For example, scholars have studied the impacts of finance [32], construction [34], and logistics [37] industries on the national economy. Third, the inter-sectoral linkages between two specific sectors attracted lots of recent research interest, for instance, real estate, construction [25], transportation, and financial services sectors [36].

The I–O model is based on an I–O table that is an inter-industry transactions table. The rows of such a table describe the distribution of a sector’s outputs throughout the economy; that sector is a supplier of other sectors. The columns describe the composition of inputs required by a particular sector to produce its outputs; that sector is a purchaser of other sectors [23].

#### 3.1.2. Linkage Measures

The essential meanings of inter-sectoral linkages are the demand and supply relationships between sectors. These linkages consist of backward and forward linkages [22,38]. Backward linkages (BLs) of a sector refer to the interdependency between the given sector and its upstream sectors from which it purchases intermediate inputs. On the other hand, forward linkages (FLs) of a sector indicate the interdependency between the given sector and its downstream sectors to which it sells its intermediate output [22].

##### Measurement of Backward Linkage

Backward linkage can be measured from the column relationship of the input–output table. The commonly used indicators are direct consumption and complete consumption coefficients, referring to direct and total backward linkages, respectively.

The direct consumption coefficient refers to the output value of other sectors needed to be directly consumed for each unit of output produced by a given sector. It can show the dependence or pulling effect of the particular sector on others:*a_ij_* = *x_ij_/x_j_*(1)
in which *a_ij_* is the direct consumption coefficient of sector *j* for sector *i*, *x_ij_* is the direct consumption of sector *i* in the production process of sector *j*, and *x_j_* is the total output of sector *j*. The economic implication of *a_ij_* is the direct consumption of the products by sector *i* when producing a one-unit output of sector *j*.

The complete consumption coefficient measures the complete dependency relationship between two sectors, including direct and indirect consumption. The calculation of the complete consumption coefficient is based on the calculation of the direct consumption coefficient:*B* = *(I* − *A)*^−1^ − *I*(2)
where *B* is the complete consumption coefficient matrix, *A* refers to the direct consumption coefficient matrix {*a_ij_*}, and *I* is the identity matrix. *(I* − *A)*^−1^ is called the Leontief inverse matrix. The larger the direct or complete consumption coefficient is, the greater the dependency of sector *j* on sector *i*.

Another index for measuring backward linkage is called the power of dispersion coefficient, which is defined as the column sum divided by the global average of the Leontief inverse matrix:(3)$$PD_{j}=\overset{n}{\underset{i=1}{\sum }}{\bar{b}}_{ij}/\frac{1}{n}\overset{n}{\underset{j=1}{\sum }}\overset{n}{\underset{i=1}{\sum }}{\bar{b}}_{ij}$$
in which *PD_j_* refers to the power of the dispersion coefficient of sector *j*, ${\bar{b}}_{ij}$ refers to the element of the Leontief inverse matrix, and *n* refers to the number of sectors in the given I–O table. If *PD_j_* is larger than 1.0, it indicates that a unit change in the final demand of sector *j* would create an above-average increase in the whole economy’s activity.

##### Measurement of Forward Linkage

Forward linkage can be measured from the row relationship of the input–output table. The commonly used indicators are direct allocation and complete allocation coefficients, referring to direct and total forward linkages, respectively.

The direct allocation coefficient reflects the use of a sector’s output as intermediate products. In other words, it refers to the proportion of a sector’s output flowing to other sector to the given sector’s total outputs:*r_ij_ = x_ij_/(x_i_* + *M_i_)*(4)
where *r_ij_* is the direct allocation coefficient of sector *i* for sector *j*, *x_ij_* is the direct intermediate output of sector *i* to sector *j*, *x_i_* is the total output of sector *i*, and *M_i_* is the imports of sector *i*.

The complete allocation coefficient reflects the total usage destination of a sector’s products, including a direct and indirect destination:*W* = (*I* − *R*)^−1^ − *I*(5)
where *W* is the complete allocation coefficient matrix, *R* is the direct allocation coefficient matrix {*r_ij_*}, and *I* is the identity matrix. *(I* − *R)*^−1^ is called the Ghosh inverse matrix. The larger the direct or complete allocation coefficient, the greater the pushing effect of sector *i* on sector *j*.

Another index for measuring forward linkage is the sensitivity of dispersion coefficient, which is defined as the row sum divided by the global average of the Ghosh inverse matrix:(6)$$SD_{i}=\overset{n}{\underset{j=1}{\sum }}{\bar{r}}_{ij}/\frac{1}{n}\overset{n}{\underset{i=1}{\sum }}\overset{n}{\underset{j=1}{\sum }}{\bar{r}}_{ij}$$
in which *SD_i_* refers to the sensitivity of the dispersion coefficient of sector *j*, ${\bar{r}}_{ij}$ refers to the element of the Ghosh inverse matrix, and *n* refers to the number of sectors in the given I–O table. If *SD_j_* is larger than 1.0, it indicates that a unit change in the final demand of all sectors would create an above-average production increase in sector *i*.

Table 1 shows the various linkage measures and their equations.

Various combinations of power and sensitivity of dispersion coefficients provide indicators for the in-depth understanding of a given sector in an economy. The sectors are classified into four types according to the combination of power and sensitivity of dispersion coefficients (Table 2) [22]. The value of 1.0 for the power and sensitivity of dispersion coefficients means the average effects of a given sector in simulating other sectors and being influenced by other sectors, respectively. First, the sectors with high power and sensitivity of dispersion (HH sectors) are always regarded as key sectors in a given economy, such as the transport sector [39]. They purchase abundant intermediate products from other sectors during their productive activities; meanwhile, they supply a large amount of intermediate products to other sectors’ production. These sectors would greatly impact the whole economy when changing their productive activities and final demands. Second, the sectors with high power but low sensitivity of dispersion coefficients (HL sectors) play an important role in pushing the economy, while their demand for other sectors’ products is relative low. They are dependent on the inter-sectoral supply. Third, the sectors with high sensitivity but low power of dispersion coefficients (LH sectors) serve as fundamental roles in the economy. Although their driving power on the entire economy is limited, they directly and indirectly induce other sectors by supplying a large amount of intermediate products. They are dependent on inter-sectoral demand. Agricultural sector, for instance, is a typical LH sector [40]. Fourth, the sectors with low power and sensitivity of dispersion coefficients (LL sectors) are always independent of other sectors or strongly connected to other sectors. In empirical analysis, quadrant graph is often used to represent these four types [30]. The quadrant graph is organized showing the power of dispersion on the vertical and the sensitivity of dispersion on the horizontal axis. The origin of the graph is 1.0. HH sectors can readily be identified in the upper right corner of the graph.

### 3.2. Data Sources

Original China’s I–O tables for 2002, 2005, 2007, 2010, 2012, 2015, 2017, and 2018 were released by the National Bureau of Statistics of China (data.stats.gov.cn (Accessed on: 2 February 2021)). The provincial I–O tables for 2002, 2005, 2007, 2010, 2012, 2015, and 2017 were released by the provincial bureaus of statistics. There are two ways to get the provincial data: one is to download from the websites of the provincial bureaus of statistics, and the other is to apply the data from the provincial bureaus of statistics. In our study, we analyzed two pioneering provinces of green finance practices in China, i.e., Guangdong and Zhejiang. The green loan balance in Zhejiang and Guangdong ranks the first and third among the provinces in China, accounting for 15% of the national total together. The amounts of green bonds and green insurance in the two provinces also come out on top among the provinces. Furthermore, we also considered the data availability of I–O tables. Guangdong’s I–O tables were downloaded from its bureau of statistics (stats.gd.gov.cn/trcc/index.html (Accessed on: 2 February 2021)). Zhejiang’s I–O tables were applied from its provincial bureau of statistics. There are two kinds of both national and provincial I–O tables, with 42 sectors and more than 100 sectors. The recent tables in 2017 and 2018 were 149- and 153-sector tables, and the tables in the other years were 42-sector tables.

In general, the environmental protection industry includes all economic activities that produce environmentally-oriented products and services. These products and services can be sorted by environmental pollution reduction and natural resource depletion, included in manufacturing and service industries, for example, manufacturing of environmental protection equipment, manufacturing of chemicals for environmental pollution treatment, instrumentation manufacturing of environmental monitoring, sewage treatment, and waste recovery. There is no single separate EPS in China’s I–O tables. The environmental protection activities are included in some sectors. For example, manufacturing of environmental protection equipment is included in the manufacturing of special equipment. Another example is the manufacturing of special pharmaceutical materials for environmental pollution treatment, which is included in the manufacturing of chemical raw materials and chemical products. Due to limited data, it is difficult to disaggregate all environmental protection activities from their corresponding sectors in the classical I–O table. In this study, we only focus on three sectors related to environmental protection activities, which are three separate sectors in the I–O table (Table 3).

To analyze the linkages between EPS and finance sectors, we compiled the I–O tables of EPS and finance sectors, which disaggregate environmental protection-related and finance-related sectors from the original I–O tables. The specific steps are as follows: first, three environmental protection sub-sectors were selected and aggregated into the environmental protection sector. Second, the finance sector was extracted from the service industry. Third, the other secondary sectors, except for the water production and supply and waste resource utilization sectors, were aggregated into a new secondary sector. The other tertiary sectors, except for the finance, water conservancy, and environmental and public facilities management sectors, were aggregated into a new tertiary sector. Finally, a new I–O table with five sectors was compiled, including the EPS, finance, primary, secondary, and tertiary sectors.

## 4. Results

### 4.1. Industrial Linkage Effects and Temporal Variations of the EPS

As shown in Figure 1, during 2002–2015, the EPS developed gradually from a sector with weak power of dispersion and strong sensitivity of dispersion into a strong power and weak sensitivity dispersion sector. Table 4 shows that the power of dispersion coefficient showed an increasing trend and exceeded that of the primary and tertiary industries in 2010 and 2015; this trend is consistent with the conclusions drawn by Wang [41]. The pulling effect of the EPS on the national economy exceeded the average sectoral level, generating an above-average motivation effect and an increasing trend. This could be attributed to the high intermediate input rate of the EPS, valuing from 35.3% in 2003 to 64.6% in 2015. During 2002–2015, increasing one-unit final demand of the EPS could increase above-average final demand of other sectors. However, this increasing trend ended in 2015, and the power of dispersion coefficient decreased to the below-average level (less than 1.0) in 2017 and 2018. The intermediate input rates of the EPS in 2017 and 2018 also decreased to 42.1% and 39.4%, respectively.

Table 5 lists the sensitivity of the dispersion coefficients of the various sectors. The EPS had fluctuant sensitivity of dispersion coefficients before 2010, with the largest value of 1.250 in 2007 and the smallest value of 0.858 in 2010; however, the EPS had values around the average level (1.0). The variation after 2010 indicated that the EPS had an average constraint on the economic growth in all sectors. When the other sectors increased one-unit demand, the products of the EPS would increase by about one unit.

Regarding the sub-sectors, the *waste resource utilization sub-sector* had obvious differences in changing trends with the other two sub-sectors (Table A1 and Figure A1). Specifically, both the *EP service sub-sector* (except 2015) and the *water production and supply sub-sector* had above-average power of dispersion coefficients with values larger than 1.0; while the *waste resource utilization sub-sector* had values ranging from 0.472 to 0.678, but in 2010 and 2015, it suddenly increased up to 1.0. On the other hand, for the sensitivity of the dispersion coefficients, the *EP service sub-sector* had values ranging from 0.531 to 0.739, with small fluctuations. In the *water production and supply sub-sector*, the coefficients dropped from 1.117 in 2002 to 0.850 in 2018. *The waste resource utilization sub-sector* had an above-average level, with values larger than 1.0 and large fluctuations. In most years, the *EP service sub-sector* was the HL sector, which had above-average increases in the activity of the whole economy when it had a unit change in final demand. The *waste resource utilization sub-sector* was the LH sector, with a production that was sensitively influenced by other sectors’ changes in final demand.

Regarding the two provinces studied, the evolution of the EPS’s industrial linkages presented differently with the national evolution (Figure 1 and Table 6). In the Guangdong province, the EPS became a key sector, with high power and sensitivity of dispersion during 2010 to 2015. However, the EPS changed to the LH sector, which was dependent on inter-sectoral demand. In the Zhejiang province, the EPS became a key sector during 2007 to 2015, a little earlier than that in Guangdong. In 2017, Zhejiang’s EPS became an LL sector.

### 4.2. Industrial Linkage between the EPS and Finance Sectors

#### 4.2.1. Backward Linkage Analysis

From the perspective of backward linkage analysis, the finance sector is the upstream industry of the EPS, and provides intermediate inputs for EPS production. From the perspective of green finance, the more those inputs flow from the finance sector to the EPS, the better the development level of green finance. Considering the measurement indices of linkage analysis, the larger the direct and total consumption coefficients of the EPS to the finance sector, the greater the strength of linkage between both sectors.

The direct and total sectoral linkages from the finance sector to the EPS in the country and the two provinces are shown in Figure 2a, using direct and complete consumption coefficients, respectively. With values between 0.007 and 0.07, the direct purchases of the EPS from the finance sector per monetary unit are reflected. In other words, the direct inputs from the finance sector were worth between 7 and 70 Yuan when the EPS produced products worth 1000 Yuan. The finance sector of Guangdong in 2007 had the greatest direct input to the EPS, and the finance sector of the country in 2002 had the lowest direct input to the EPS. A trend of rising first and then decline was apparent for the country and the Zhejiang province. The inflection points were both in 2015. However, Guangdong’s trend rose before 2007, then declined and then rose again in 2017.

The total sectoral linkages from the finance sector to the EPS varied between 0.024 and 0.140. The finance sectors of Guangdong in 2010 and the country in 2002 had the highest and lowest total effect on the EPS, respectively. Similar trends to the direct linkages occurred in Zhejiang and the country, rising before 2015 and then falling. In Guangdong, the increasing trend occurred between 2002 and 2010, the declining trend between 2010 and 2012, and a stable trend between 2012 and 2017.

#### 4.2.2. Forward Linkage Analysis

From the perspective of forward linkage analysis, the EPS is taken as the downstream sector of the finance sector, and the supply strength of the finance industry on the EPS is examined. Among the products produced by the finance sector, if the proportion of inputs toward the EPS is large, it indicates that the finance sector’s support to the EPS is large. The larger the direct and complete allocation coefficients of the finance industry to the EPS, the stronger the finance industry’s pushing effect is on the EPS.

Figure 2b shows the direct and total sectoral linkages from the finance sector to the EPS. The values stabilized between 0.003 and 0.011 in Zhejiang and the country, indicating the small strength of the forward linkages between the finance sector and EPS sector. Zhejiang and the country both had rose first and then decreased in the direct allocation coefficients’ evolutions, with the inflection points both in 2015. In Guangdong, the direct allocation coefficients were larger than the former regions, ranging from 0.010 and 0.019. Guangdong had a trend of rising before 2007, decline before 2012, and then a little rising between 2012 and 2017. Regarding the total allocation coefficients, the values ranged from 0.009 and 0.049. The highest value was in Guangdong in 2010, and the lowest value was the national average in 2002. The results indicate that Guangdong’s finance sector provided a relatively larger and earlier support to the local EPS. It could be explained that Guangdong is one of the seven pilot carbon markets, which began in 2011. The trading amount of carbon emission quotas in Guangdong was the largest in the seven pilot markets.

To summarize, the results indicate different trends in the role of the EPS at the national and provincial levels. However, the common situation is the importance of the EPS in the economy and that the supports from the finance sector both decreased after 2015. The potential explanation and policy implications will be addressed in the following section.

## 5. Discussion and Policy Implications

### 5.1. On the Support of the Finance Sector to the EPS

The backward linkage analysis shows that the national average relationship between the finance and EPS sectors increased gradually, reached its peak in 2015, and then declined. In the Zhejiang province, the relationship between both sectors had a similar changing trend as the national average. In the Guangdong province, the peaks occurred earlier than the national average. The backward linkages between the two specific sectors demonstrated a significant upward trend from 2010 to 2015. The direct and complete consumption coefficients of the EPS and finance sectors indicated significant increases, indicating that the EPS increasingly relied on the finance sector. This indication was consistent with the support and policy orientation of China in recent years toward the development of green finance, and could prove the effectiveness of the development of green finance.

The forward linkages between the two specific sectors showed a fluctuation changing trend during the study period at both the national and provincial levels. There were two peaks in 2010 and 2015, but the general trends decreased. In the unit product of the finance sector, the proportion of inflow to the EPS was relatively smaller than that of other sectors. This can be explained that the three environmental protection sub-sectors produce public goods or quasi-public goods, which have a small attraction to financial investment. Investors have a strong interest in developing other environmentally friendly industries, like sustainable energy production, improvement of environmental protection facilities, etc. These kinds of environmentally friendly industries have more economic return than the three environmental protection sub-sectors.

Therefore, we proposed some policy implications in promoting financial service support in the EPS. The efficient mechanism for linking the finance and EPS sectors should be improved further.

(1) Green finance approaches should be innovated; these innovations may include the development of a green industry fund, the establishment of specialized green banks, improvement of the statistical system for green credit, and development of green finance evaluation systems.

(2) Innovative green finance products should be encouraged to design and release. For example, the formation of an environmental rights market makes environmental rights become collateral for business loans; financial institutions can develop new financial products based on environmental rights, such as carbon emission rights, emission rights, and energy use rights.

(3) Environmental risk management in financial institutions should be emphasized. Institutional investors should understand novel risk sources, which are caused by tightened environmental and climate policy and frequent environmental and climate damage. To prevent investments from these environmental risks has been a new challenge for financial practitioners.

### 5.2. On the Development of the EPS

The temporal analysis demonstrated that the EPS has gradually developed from a sector with weak backward and strong forward linkages to a sector with strong backward and weak forward linkages from 2002 to 2015. The power of dispersion exhibited above-average influence on other sectors, meaning that a unit change in the final demand of the EPS created an above-average increase in the activity of the whole economy. The relationship between the EPS and the national economy has become increasingly more robust. However, in the recent years of 2015 to 2018, it returned to a sector with weak power of dispersion and strong sensitivity of dispersion. Therefore, the promotion of the development of the EPS should be encouraged. Its development showed not only the need for sustainable development, but it also stimulated economic development. China should continue strengthening EPS development and promoting its pulling effect on other sectors.

In the final consumption structure of the EPS, government and household consumption accounted for approximately 70% and 30%, respectively. Although household consumption accounted for a small portion, it showed an increasing trend in recent years. Therefore, household participation and consumption for the EPS should be increased to improve public participation. Approaches should include establishing private green financial institutions, innovation of private green products (e.g., a green credit card) and businesses (e.g., individual carbon credit by Ant Financial Services Group), and promotion of public participation in green finance.

### 5.3. On the Development of Green Finance at the Provincial Level

Zhejiang and Guangdong are the pioneering provinces of green finance practices. The EPS in the two provinces has become well-developed in recent years. For example, the two provinces are the top two in enterprise number and annual income of enterprise in the EP service sub-sector, which have gathered more than 20% of the enterprises and created about one-quarter of the country’s annual income. However, the EPS in the two provinces have become weakly connected to other sectors in recent years.

The linkages between the EPS and finance sectors in Guangdong presented a pioneering trend; that is, the trend rose first, then declined, and then rose in the recent years. Therefore, Guangdong’s experience can be extended to other provinces to promote finance support in the development of EPS. It should be helpful further to study Guangdong’s experience in detail. For example, the environmental rights trading market is developing rapidly. Guangdong has the country’s largest carbon trading market. Moreover, in Guangdong, cultivating talents who are familiar with the operation of the capital market and the characteristics of green industries is important for financial institutions. Through selection, prize awarding, and promotion of excellent green finance cases, financial institutions encourage innovative green financial practices and expand their influence.

## 6. Conclusions

This study provides a methodology framework to evaluate the support from the finance sector to the environmental protection sector at both national and provincial levels. Specifically, we use linkage analysis based on the input–output model to measure the connection between two specific sectors. We also analyze the changes of national and provincial roles of the EPS from 2002 to 2017. The results help us to understand the development of green finance from the perspective of sectoral relationships.

This study has shown that the EPS in China has gradually developed from a sector with weak backward and strong forward linkages to a sector with strong backward and weak forward linkages from 2002 to 2015; however, in 2017 and 2018, the EPS returned to a sector with weak backward and strong forward linkages. In the two studied provinces, the EPS used to involve sectors with strong backward and strong forward linkages, which indicated that the EPS was a key sector at the provincial level. Another significant finding is that the I–O relationship between the EPS and finance sectors rose first and then declined in the country and in the Zhejiang province; Guangdong had a similar evolution in the former period, but it had a rising trend in the I–O relationship between the EPS and finance sectors in the latest year. The findings of this research provide insights for further promoting support from the finance sector to environmental protection activities.

This study, however, has some limitations. First, the development of the EPS is unequal in different regions of China. For example, the four regions—namely Beijing, Zhejiang, Guangdong, and Jiangsu—account for close to 52% of the operating revenue of the national EPS [3]. It is necessary to analyze the linkages between the EPS and finance industries in more provinces and compare the results. However, in this study, only two provinces were studied due to data limitations. Second, because the I–O table has limitations on sector disaggregation, this study only reflects the situation of three sub-sectors of the environmental protection industry. More sub-sectors are included in manufacturing sectors, such as manufacturing of environmental protection equipment, manufacturing of chemicals for environmental pollution treatment, and instrumentation manufacturing of environmental monitoring. Based on more detailed data, the linkages between the more complete environmental protection sector and finance sector can be calculated, and then the assessment on green finance development can be updated.

Future research in the following fields would be of great help to understand green finance more deeply. To analyze the linkages in more places, including provinces or cities, is needed to explore to what extent the green finance develops in different areas. Moreover, the results of the linkages in various places could be used to relate with other economic indices, such as the GDP, fixed investment, or government subsidies. In that case, the driving forces of developing green finance could be further explored. Regarding a broader definition of green finance, economic activities regarding carbon emission mitigation, environmental prolusion decreases, and natural resource saving can be included. For example, renewable energy productions (such as wind power and solar power generation) reduce carbon emissions significantly, and are regarded as major investment targets of green finance. The renewable energy productions should be disaggregated from the electricity generation sector based on some detailed wind and solar power generation data. In the future study, the method in this study can be extended to calculate financial support on all green finance activities based on more detailed data.

## Figures and Tables

**Figure 1 ijerph-18-02634-f001:**
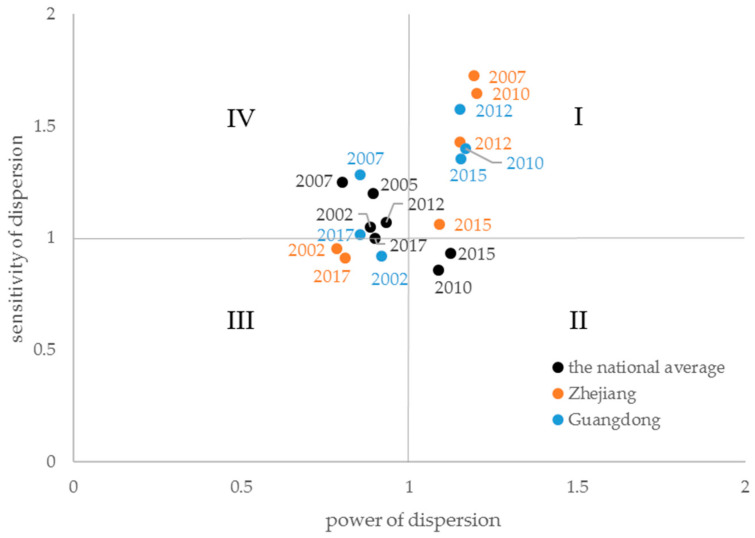
Power and sensitivity of dispersion coefficients of the EPS during the study period. The figures next to the dots are the year of the corresponding dots.

**Figure 2 ijerph-18-02634-f002:**
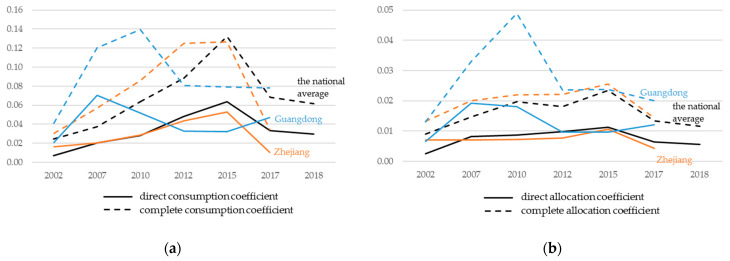
Trends of the industrial linkage coefficients of the national average, Guangdong, and Zhejiang between the finance and environmental protection sectors: (**a**) direct and complete consumption coefficients; (**b**) direct and complete allocation coefficients.

**Table 1 ijerph-18-02634-t001:** Linkage measures and their equations.

Industrial Linkage	Direct Linkages	Total Linkages	Normalized Linkages
Backward linkages	Equation (1)	Equation (2)	Equation (3) ^1^
Forward linkages	Equation (4)	Equation (5)	Equation (6) ^2^

^1^ The normalized backward linkages are also called the power of dispersion coefficients; ^2^ the normalized forward linkages are also called the sensitivity of dispersion coefficients.

**Table 2 ijerph-18-02634-t002:** Four types of sectors with the combination of power and sensitivity of dispersion coefficients.

Classification Value	Sensitivity of Dispersion < 1	Sensitivity of Dispersion > 1
Power of dispersion <1	Generally independent (LL)	Dependent on inter-sectoral demand (LH)
Power of dispersion >1	Dependent on inter-sectoral supply (HL)	Generally dependent (HH)

**Table 3 ijerph-18-02634-t003:** EPS and finance sectors in the original I–O table and the established I–O table.

Sectors in the Original I–O Table		Sectors in the Established I–O Table
42-Sector I–O Table	More Than 100-Sector I–O Table	
Waste resource utilization (in the secondary industry)		Environmental protection sector (EPS)
Water production and supply (in the secondary industry)		
Water conservancy and environmental and public facility management (in the tertiary industry) ^1^	Water conservancy management	
	Ecological protection and environmental management	
	Public facility management	
Finance (in the tertiary industry)	Monetary finance service	Finance sector
	Capital market service	
	Insurance service	

^1^ It is called the *EP service sub-sector* for short.

**Table 4 ijerph-18-02634-t004:** Power of dispersion coefficients of various sectors.

Industry	2002	2005	2007	2010	2012	2015	2017	2018
EPS	0.885	0.892	0.802	1.087	0.931	1.124	0.897	0.876
Finance	1.049	1.198	1.250	0.858	1.068	0.932	0.998	1.045
Primary	0.840	0.832	0.750	0.722	0.809	0.708	0.855	0.873
Secondary	1.168	1.003	0.988	1.087	1.079	1.086	1.061	1.035
Tertiary	0.917	0.873	0.925	0.864	0.896	0.867	0.899	0.882

**Table 5 ijerph-18-02634-t005:** Sensitivity of dispersion coefficients of various sectors.

Industry	2002	2005	2007	2010	2012	2015	2017	2018
EPS	1.049	1.198	1.250	0.858	1.068	0.932	0.998	1.045
Finance	0.840	0.832	0.750	0.722	0.809	0.708	0.855	0.873
Primary	1.168	1.003	0.988	1.087	1.079	1.086	1.061	1.035
Secondary	0.917	0.873	0.925	0.864	0.896	0.867	0.899	0.882
Tertiary	0.934	1.001	1.004	1.157	1.055	1.148	1.119	1.093

**Table 6 ijerph-18-02634-t006:** Power and sensitivity of dispersion coefficients of the EPS in Zhejiang and Guangdong.

Province	Dispersion Coefficient	2002	2007	2010	2012	2015	2017
Zhejiang	Power	0.784	1.192	1.201	1.150	1.090	0.808
	Sensitivity	0.951	1.725	1.648	1.428	1.062	0.911
Guangdong	Power	0.916	0.855	1.168	1.152	1.155	0.854
	Sensitivity	0.918	1.282	1.398	1.574	1.356	1.017

## Data Availability

The data used are all public data which can be download in the websites mentioned in the paper. The authors will also provide the data if the interested readers require them.

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
