# Peer review of "Linkage Analysis between Finance and Environmental Protection Sectors in China: An Approach to Evaluating Green Finance"

_ijerph, 2021, doi:10.3390/ijerph18052634_

Round 1

Reviewer 1 Report

Dear authors, 

Thank you for your interesting analysis. Below some minor elements: 

1) I suggest to create a section dedicate to the literature review including several elements now present in the introduction section; 

2) the gap of the paper is now fully addressed. Therefore, I would suggest to strengthen it. 

In the literature review, you should consider as suggested in the first part of the paper the need (climate change) and then create a dedicate sub-paragraph, including green finance as a possible answer to it. 

Therefore --> 1. Introduction --> 2. Literature Review - 2.1 Climate change 2.2. Green finance 

In the introduction, you should clarify: 

  • the background of the paper; 
  • the main GAP that your paper wants to address; 
  • what is the research question(s) that you address?
  • the main theoretical and practical implications of your paper
  • the map of the paper. 

You may consider the following papers: 

Milne, M. J., & Grubnic, S. (2011). Climate change accounting research: keeping it interesting and different. Accounting, Auditing & Accountability Journal.

Lovell, H., & MacKenzie, D. (2011). Accounting for carbon: the role of accounting professional organisations in governing climate change. Antipode, 43(3), 704-730.

Secinaro, S., Brescia, V., Calandra, D., & Saiti, B. (2020). Impact of climate change mitigation policies on corporate financial performance: Evidence‐based on European publicly listed firms. Corporate Social Responsibility and Environmental Management, 27(6), 2491-2501.

Berensmann, K., Volz, U., Alloisio, I., Bak, C., Bhattacharya, A., Leipold, G., ... & Yang, Q. (2017). Fostering sustainable global growth through green finance–what role for the G20. T20 Task Force on Climate Policy and Finance.

D’Orazio, P., & Popoyan, L. (2019). Fostering green investments and tackling climate-related financial risks: Which role for macroprudential policies?. Ecological Economics, 160, 25-37.

Methodology

The method is well-written and interesting. 

Results

Results are fully interesting for the reader. 

Discussion

I would also suggest considering the better theoretical implications of your study. 

Conclusions

Please, add limitations and future research challenges for future researchers.

All the best,

The Reviewer 

Reviewer 2 Report

The paper has a very interesting perspective and also a very interesting test. I like it. Despite this, some issues must be improved. I made these suggestions to enhance the paper’s quality and to ensure a higher impact than the one expected in the current version.

Even if most of the mandatory improvements or suggestions could be considered “minor changes”, suggestion 5 makes me recommend “major changes” in my review.

I present you the suggested improvements next:

  1. It could be of interest if you give more detail about the environmental industry. What it is? How does it work? How could it be of benefit? What is different from other “common” industries? This could be of use for the unrelated or introductory reader. Also for other environmental, social responsibility, or sustainability research areas.
  2. The literature quoted in lines 72 to 81 is very interesting and appropriate for the introduction but your brief introductory description does not say the places in which these were assessed. Even if you describe these in detail in the previous literature section, it could be of use and interest to mention the places in which these research works were developed.
  3. In lines 138 to 146, the authors properly explain the gap in the literature that they want to fill. It is appropriate the way they present their writing but needs more “punch” (sorry for the conventional language, but I like your paper). Please enhance the writing style to be more emphatic in the gap that you fill. Also, if it is possible and the case, the difference between your quantitative method and others and the improvements made by your research. This is one of the key paragraphs of your paper in which you emphasize the rationale and novelty of your research work.
  4. Do you manage a hypothesis or research aim? It is obvious you do it, but if, after line 141, you present a hypothesis or research aim, you will strengthen your exposition and highlight your contribution.
  5. You need a literature review of previous works. The authors are well aware that it is still the common practice to present a review of the previous works that motivate the current one. It is important to know what do these works do, how do they do it, why do they do it (if possible) and what are their limitations and motivations of their results to the current work. Please, present it after the introduction. We want to know with more detail the background of your research and the extensions made with it.
  6. In line 254 please present the proper quote of the National Bureau of Statistics of China as a website quote. More specifically, with the link of the page in which the interested reader can download the data for replication. The same recommendation must be made for the provincial bureaus of statistics.
  7. Also, it is necessary to explain briefly why are you analyzing these provinces (you mention it until line 430. Too late.).
  8. Also, you haven’t tell what software or programming language you are using for your calculations. If it is possible, please add the input data that you processed as supplementary materials and, if you prefer it, also the analysis code or file (this is optional or you can mention that it is available upon request to the corresponding author). I suggest this because, as noted from Table 3 and its motivations, you made a lot of hard and interesting input data processing that could be of use for potential extensions and replication of your work. If you finally decide to provide all this as supplementary material and are using R or Python, please write a Rmarkdown or Jupyter notebook. This notebook could be of use to the final reader of your paper. As I told you all this is optional but could be of use for the interested reader.
  9. Line 292: “However, this increasing trend terminated in 2015”. I suggest ended instead of terminated.
  10. Please, give a potential explanation of your results. E. g. In Guangdong, the direct allocation coefficients were larger than the former regions, ranging from 0.010 and 0.019. Please give a potential discussion or explanation of why in this and all the results. That’s part of your research grounds and could be of introductory use for section 4 “Discussion and policy implications”.
  11. A corollary of results before section 4 could be useful for the reader. This, to give a general perspective (a summary) of your findings. A useful summary for your next section.
  12. Please check line 380. There are some unnecessary dashes.
  13. Research question: Is it possible to develop an unbalanced panel data analysis (you have some missing years in your provinces)? An analysis in which you relate the EP investment with the linkages. Does your input and output analysis data give the potential to do this? If so, it could be an interesting analysis that could strengthen your results and policy recommendations. This is optional to do it. If you don’t do it, please at least mention it either in your work’s limitations or your guidelines for further research.
  14. The three recommendations or lines 395 to 408 must be numbered and not as part of the paragraph. This is one of the most important parts of your paper. Please highlight your recommendations.
  15. Line 458: In two studied provinces. I suggest “In the two studied provinces”.
  16. In the conclusions section, please mention briefly your policy recommendations, limitations, and guidelines for further research (at least two or three of these).
  17. The writing style is good but needs some marginal improvements. Please send your paper’s final version to a third-party reviewer (if possible, present the certificate to the editor). I suggest this to attend small potential issues left unattended.
  18. Please, in your answer letter highlight your changes with orange text to identify them in the paper’s new version.

Good work. Congratulations.

Reviewer 3 Report

Abstract: 
Well presented, with all the necessary information to realize the objective of the study. I have nothing to propose for alteration.

Introduction:
The authors have chosen to join the introduction and the literature review in the same section. Personally I like to divide these sections but the way the authors presented this section is very well done. Nothing to add. Very good.

Methodology:
In this section I would like to see cited some studies that have used this same methodology. These references are important to make the study more robust through the previous background. You could create a short section with 
a) Previous Studies using these methods
It's just an idea that came to me from reading. Although I consider your methodology very complete.

Results:
I would like to see the statistical significance of your data introduced.
What robustness tests have you done? What are the results of these tests?
It is essential to demonstrate the validity and reliability of the data entered. This will make the research work more robust.

Conclusions:
At the end of this conclusion, the authors should introduce at least one or two paragraphs listing the limitations of this study and proposals for future research. Such information is important for these or other authors to be able to follow up on this type of studies or related studies.

Final Notes: I have enjoyed reading this article. The topic is interesting and emerging. 
After small corrections and improvements I will be able to give a favourable opinion for the publication.
I wish the authors all the best and much health in these difficult times.
The reviewer

Reviewer 4 Report

Interesting subject, but lacking a broader context. This requires discussion and improvement. 

The authors should explain in detail why the Chinese case is so important. This cannot be a type explanation because China is important, it is a big economy, etc. A scientific explanation must be provided. 

Tthe reviewer understand the authors' choice of method. The reviewer learned about the need for an alternative approach. However, no explanation is given as to why other metors have not been chosen, whether they exist, whether they can be used, but the only method is using the linkage analysis method based on the new tables at the national level. 

Very poor selection of literature. This needs to be completed as there are other studies, other authors who have achievements. On the basis of the (broad) discussion, it is only possible to embed and discuss the achievements of Chinese or Asian authors. 

Very poor conclusions. There is no reference to the theory in economics and finance, just as there is no basis in literature review. There is no link with future research intentions. 

Round 2

Reviewer 2 Report

None, they attended all the changes.

Reviewer 4 Report

I am satisfied with the revision of the article